# Healthcare Equity and Commissioning: A Four-Year National Analysis of Portuguese Primary Healthcare Units

**DOI:** 10.3390/ijerph192214819

**Published:** 2022-11-10

**Authors:** António Pereira, André Biscaia, Isis Calado, Alberto Freitas, Andreia Costa, Anabela Coelho

**Affiliations:** 1Family Health Unit, Unidade de Saúde Familiar Prelada, ACES Porto Ocidental, 4250-113 Porto, Portugal; 2PHC—Commissioning Department, Northern Regional Administration of Health, 4000-447 Porto, Portugal; 3CINTESIS—Center for Health Technology and Services Research, Faculty of Medicine, University of Porto, 4200-319 Porto, Portugal; 4Family Health Unit, Unidade de Saúde Familiar Marginal, ACES Cascais, ARS Lisboa e Vale do Tejo, 2765-618 São João do Estoril, Portugal; 5University College London Medical School, London WC1E 6DE, UK; 6Nursing Research, Innovation and Development Centre of Lisbon (CIDNUR), Nursing School of Lisbon (ESEL), 1600-096 Lisbon, Portugal; 7Católica Research Centre for Psychological, Family and Social Wellbeing, Faculdade de Ciências Humanas, Universidade Católica Portuguesa, 1649-023 Lisbon, Portugal; 8Instituto de Saúde Ambiental (ISAMB), Faculdade de Medicina, Universidade de Lisboa, 1099-085 Lisbon, Portugal; 9Comprehensive Health Research Centre (CHRC), Nursing Department, University of Évora, 7004-516 Evora, Portugal; 10H&TRC—Health & Technology Research Center, ESTeSL—Escola Superior de Tecnologia da Saúde, Instituto Politécnico de Lisboa, 1990-096 Lisbon, Portugal; 11Global Health and Tropical Medicine, Instituto de Higiene e Medicina Tropical, Universidade NOVA de Lisboa, 1349-008 Lisbon, Portugal

**Keywords:** primary healthcare, health equity, commissioning, family health units, community health, health policy, socio-economic factors, community-based health financing, community health services

## Abstract

Equal and adequate access to healthcare is one of the pillars of Portuguese health policy. Despite the controversy over commissioning processes’ contribution to equity in health, this article aims to clarify the relationship between socio-economic factors and the results of primary healthcare (PHC) commissioning indicators through an analysis of four years of data from all PHC units in Portugal. The factor that presents a statistically significant relationship with a greater number of indicators is the organizational model. Since the reform of PHC services in 2005, a new type of unit was introduced: the family health unit (USF). At the time of the study, these units covered 58.1% of the population and achieved better indicator results. In most cases, the evolution of the results achieved by commissioning seems to be similar in different analyzed contexts. Nevertheless, the percentage of patients of a non-Portuguese nationality and the population density were analyzed, and a widening of discrepancies was observed in 23.3% of the cases. The commissioning indicators were statistically related to the studied context factors, and some of these, such as the nurse home visits indicator, are more sensitive to context than others. There is no evidence that the best results were achieved at the expense of worse healthcare being offered to vulnerable populations, and there was no association with a reduction in inequalities in healthcare. It would be valuable if the Portuguese Government could stimulate the increase in the number of working USFs, especially in low-density areas, considering that they can achieve better results with lower costs for medicines and diagnostic tests.

## 1. Introduction

### 1.1. Health Equity in the Portuguese Context

Portugal has marked socio-economic inequalities, having one of the most unequal income distributions in Europe [1]. The Portuguese Constitution (1976) outlines that the state should, “ensure all citizens have access to preventive, curative and rehabilitative healthcare services, regardless of their socio-economic condition” [2], and this commitment is strengthened by the 1990 Health Act, which declares that ensuring equity in the distribution of resources and the use of services is a fundamental goal of the country’s health system [3]. Despite this, socio-economic disparities continuously translate into health inequalities. Determinants such as low income, lower educational attainment, female gender and migrant status have been linked to worse health outcomes in conditions such as obesity [4,5,6,7], mental health problems [8,9] and cardiovascular disease [10,11]. In addition, Portugal presents disparities between the rich and the poor in the ability to access both specialist and primary healthcare consultations [12,13]. These disparities are also seen in ambulatory care-sensitive conditions, suggesting worrisome inequalities in the access to early and high-quality primary healthcare [14].

The concept of health equity was brought into the mainstream of national health policy through the 2015–2020 National Health Plan [15], following a WHO evaluation that identified a concerning policy gap in the area of health inequalities in previous plans [16]. The 2015–2020 plan defines equitable and adequate access of healthcare as one of the four cornerstones of the national health policy, proposing, among other strategies, the articulation of national and local health policy through commissioning processes in primary healthcare (PHC) [15].

### 1.2. The Portuguese Primary Healthcare System

In 2005, the Portuguese Government launched a country-wide reform of primary healthcare services, introducing a new type of PHC unit, the family health unit (“USFs”—Unidades de Saúde Familiar). USFs are public, self-managed primary healthcare units formed by a self-selected group of general practitioners (GPs), PHC nurses and administrative staff, with functional and technical autonomy to organize the delivery of health services [17]. All USFs can access the government’s PHC incentive scheme, which rewards good practices with grants that may be used to fund training and research activities for the PHC teams. USFs are further divided into two organizational models, A and B, where model B USFs have an additional pay-for-performance scheme to financially reward individual professionals according to workload and their team’s performance, a design that rewards quality of care and promotes teamwork [18].

The PHC reforms in Portugal can be also analyzed within their organizational framework. Before PHC reform, the predominant model was the classical management model of control with the modest involvement of professionals, based on a bureaucratic model with a hierarchical structure [19]. With the creation of the USF, a participatory management model is proposed that extensively involves all professionals (general practitioners, nurses and administrative staff) and has more organizational autonomy with the purpose of motivating the group to achieve common and commissioned goals [20]. Total quality management [21] and clinical governance [22] also contribute to the organizational model of the USF, involving several professionals in the pursuit for better quality. The three organizational models that currently coexist in Portugal (Table 1: Cf organizational models/type of unit) have different degrees of organizational maturation: UCSP is an organizational model with less autonomy that is closer to the existing model before the reform; the USF A model is an autonomous model in an early stage of organizational maturation (a model of learning and preparation) and a fixed salary; and the USF B model has the same degree of autonomy as model A, but is more mature and has a payment scheme sensitive to workload and performance.

In December 2016, 41.9% of the Portuguese population were enrolled in UCSP, 27% in USF model A and 31.1% in USF model B [23].

The 2005 reform also relaunched the commissioning processes, which are used to plan and deliver services for all types of units in the country. Our analysis of all PHC units in Portugal during the period of 2013–2016 found that, after the 2005 PHC reform, the quality indicators included in the commissioning process improved. This improvement was not associated with a detrimental effect on noncommissioned indicators and there was a general improvement in the quality of PHC services [17].

### 1.3. Commissioning and Equity: Theory and Evidence

Commissioning is formally defined as a process of procuring health services based on the assessment of the population’s needs [24]. On a practical level, it creates a separation between the provider and purchaser of services [25], which improves priority settings and service integration for population groups in need [26], thus optimizing the way services are delivered and contributing to health equity [27]. However, commissioning processes may also generate a conflict between the goals of efficiency and equity, such that more cost-effective services are procured, jeopardizing the services required by vulnerable populations [28].

The available evidence on commissioning has increased in recent years, as more countries implement such processes. Nonetheless, the extent to which these processes contribute to health equity is controversial. A review of 27 studies on the effect of the UK’s commissioning scheme, the Quality of Outcomes Framework (QOF), concluded that the scheme is overall beneficial to the improvement of equity in treatment access and intermediate treatment outcomes, but that the extent to which different patient groups benefit from this improvement highly depends on the quality indicators and service users under study [29]. Although some of the studies included in the review demonstrated a clear reduction in the attainment gap between socially deprived and advantaged groups in areas such as blood pressure monitoring [30], others highlighted how some groups of patients remain at a disadvantage despite improvements in other groups [31]. Another study assessing the general achievement of quality indicators for 7637 UK primary healthcare practices and found that the introduction of financial incentive schemes led to a significant decrease in disparities in the delivery of PHC services related to area deprivation over a period of 3 years [32]. Similarly, a recent study from Brazil also concluded that the introduction of a pilot quality improvement commissioning process in over 13,934 health teams successfully eliminated income inequalities in the delivery of PHC services [33].

The present study aims to explore the relationship between the commissioning processes and health equity in Portuguese primary healthcare services, with two main objectives: (1) to understand whether the evolution in primary healthcare quality indicators is associated with specific socio-economic context factors in which PHC units are inserted; and (2) to understand if the introduction of commissioning processes leads to an attenuation in the outcome disparities between primary healthcare units in the most and least deprived contexts.

## 2. Materials and Methods

This study used the results of PHC performance indicators from units across mainland Portugal published by the Portuguese Central Administration of the Health System (ACSS—Administração Central do Sistema de Saúde) from 2013 to 2016. This includes data from the period where the indicators were used as targets in the commissioning process (2014–2016) and the year prior to this change (Table 1: commissioning indicators for Portuguese PHC units (2014–2016)).

Indicators were determined for every PHC unit, and each unit was characterized according to its organizational model as UCSP, USF Model A, or USF Model B.

Six socio-economic determinants (Table 1: socio-economic determinants) were also used to characterize the PHC units’ populations. The percentage of elderly patients, percentage of users with Portuguese nationality and percentage of users in economic deprivation were calculated based on the ACSS per unit considering the information of each user enrolled in this unit. The unemployment rate, population density and school dropout rate before completing mandatory education are averages for the area in which the PHC unit is inserted [34].

All units that ceased to function or changed their organizational models during the study period were excluded from the analysis. According to this criterion, of the initial 1104 PHC units, 378 were excluded. The remaining 726 units accounted for 8,519,723 users out of a total of 10,664,898 SNS users (79.9%).

A descriptive analysis was performed for each indicator by year, by model and globally.

The exploratory factor analysis (EFA) technique [35] was used to analyze the evolution of the results in different socio-economic contexts before and after commissioning. The study, for the analysis of the adequacy of the factors, used the Kaiser–Meyer–Olkin (KMO) test, whose value should be greater than 0.5, and Bartlett’s sphericity test, which indicates the adequacy of the data for a factor analysis These tests analyzed the total variance explained by the results.

Indexes of context variables were created to identify the factors underlying the context of the functional units, and these factors were used to divide the units into groups. The groups were created by dividing each factor into terciles and the number of factors were defined by Pearson’s criterion (≥80%). Bartlett’s sphericity test and the KMO test were applied.

To analyze whether the evolution of commissioning indicators is related to the PHC units’ socio-economic context variables, the generalized estimating equation model was used. Using the quasi-likelihood under the independence model criterion (QIC) and corrected quasi-likelihood under the independence model criterion (QICC) analysis, a two-point analysis (2013 and 2016) was chosen. To analyze whether there is an attenuation of differences in the results in different socio-economic contexts before and after commissioning, a factor analysis was used.

The SPSS 26 (IBM Corp. Armonk, NY, USA) software was used for data analysis.

This article was the result of a research protocol approved by the North ARS Ethics Committee (CES 4/2017).

## 3. Results

### 3.1. Relationship between Context Factors and Evolution of the PHC Indicator Results (Table 2)

The context factors that showed a statistically significant association with the evolution of the results of a larger number of indicators were:The organizational model of the units (all indicators);The percentage of users with Portuguese nationality (eight indicators);The unemployment rate, percentage of elderly and population density (six indicators).

Those with a statistically significant association with the results of a smaller number of indicators were:The school dropout rate before mandatory education is completed and the percentage of users in economic deprivation (four indicators).

The indicators with an evolution that showed a statistically significant association with a greater number of context variables were:The proportion of hypertensive patients under 65 years of age and with blood pressure below 150/90 (five context variables);The proportion of patients aged 14 or older with regular smoking habits registered (five context variables);The proportion of child-bearing age women with appropriate family planning support (five context variables).

The indicators with an evolution that showed a statistically significant association with a smaller number of context variables were:The proportion of users with DM2 with the last HbA1c recorded less than or equal to 8% (three context variables);The proportion of elderly patients not on any antianxiety, sedative or hypnotic medications (two context variables).

### 3.2. Differences in Results in Distinct Socio-Economic Contexts before and after Commissioning

The results of the Kaiser–Meyer–Olkin (KMO) test (KMO measure of sampling adequacy = 0.6) and Bartlett’s sphericity test (χ^2^ = 6206.926, *p* < 0.001) indicated that the data were suitable for factor analysis.

The study of communalities presents values higher than 0.728, indicating that all variables contained in the study are explained by the extracted components and the percentage of the total explained variance was 82%.

After factorial analysis, three factors were obtained, and each factor can be mainly explained by two variables (Figure 1):Factor 1: correlated with the variables of unemployment rate (positive) and percentage of elderly patients (negative);Factor 2: correlated with the variables of school dropout rate before mandatory education completed (positive) and percentage of users in economic deprivation (positive);Factor 3: correlated with the variables of population density (negative) and percentage of users with Portuguese nationality (positive).

Analyzing the cluster that grouped the context characteristics of “unemployment rate” and “percentage of elderly patients” (F1), the discrepancies in the results were attenuated in the “proportion of hypertensive patients under 65 years of age and blood pressure below 150/90” and “proportion of patients aged 14 or older with regular smoking habits registered” following the commissioning of these indicators in USF-A-type units (change from *p* > 0.05 in 2013 to *p* < 0.05 in 2016).

In the cluster that grouped “school dropout rate before mandatory education completed “and “percentage of users in economic deprivation” (F2), the discrepancies in the results were attenuated for the “proportion of patients with appropriate maternal health follow-up” indicator after it was commissioned, but this was only verified in USCP-type units (change from *p* > 0.05 in 2013 to *p* < 0.05 in 2016).

In the cluster that grouped “population density” and “percentage of users holding Portuguese nationality” (F3), the discrepancies in results were attenuated for the “proportion of patients aged 14 or older with regular smoking habits registered” indicator after it was commissioned, but this was only verified in USF-A-type units.

### 3.3. Analysis Based on Organizational Model

We found that, in general, USFs are implemented in places with a higher average population density, a lower rate of school dropout before the end of compulsory education, a higher unemployment rate, fewer elderly people, fewer users in economic deprivation and fewer non-Portuguese users (Table 3).

In 2016, the USFs had better results in all analyzed health indicators and had a lower average cost of expenses for medicines and diagnostic tests (Table 4).

## 4. Discussion

Previous studies showed the relationship between health outcomes and population context [36], and the 2015–2020 Portuguese National Health Plan defines equitable and adequate access to healthcare as one of its four cornerstones, proposing, among other strategies, the articulation of national and local health policy via commissioning processes in primary healthcare. However, commissioning processes may also generate a conflict between the goals of efficiency and equity, such that more cost-effective services may jeopardize the services required by vulnerable populations [37,38,39,40].

This article aims to clarify the relationship between the context and the results of commissioning indicators in PHC units in Portugal.

The results of this study show that the evolution observed in the results of the commissioning indicators is statistically related to the studied context factors, and that some PHC performance indicators are more sensitive to context than others. Furthermore, the organizational model of the unit is the factor that presents a statistically significant association with a greater number of indicators (9 out of 10 indicators). Additionally, better results were achieved by the Model B USFs, followed by Model A USFs, and then, UCSPs.

This suggests that the organizational model and payment system [41,42] may have an impact on performance. Self-selected groups with higher functional autonomy (the USFs) and, within these, those with a pay-for-performance system (Model B USFs) appear to perform better than others, reinforcing the need for further investigating the link between financial incentives and quality of care. This also highlights the importance of increasing the number of working USFs.

In the factorial analysis, three factors were obtained. Factor (F) 1 could be linked to the active population and employment as it is positively correlated with the variables “unemployment rate” and negatively correlated with the “percentage of the elderly population”. F2 relates to the poverty cycle as it is positively correlated with the variables “school dropout rate before mandatory education completed” and “percentage of users in economic deprivation”. F3 is linked to territorial attractiveness, as it is negatively correlated with the variables “Population density” and positively with the “percentage of users holding Portuguese nationality”.

Overall, the differences or similarities existing before commissioning remain unchanged in more than 80% of cases, but it is important to note that, in F1, the absence of an association between the commissioning process and the results obtained by the PHC units is observed in 90% of cases; in F2, this value reaches 80%, and in F3, it is 63.3%. This means that commissioning may have a greater impact on (both improving and worsening) the performance of PHC quality indicators in contexts with a higher percentage of patients of non-Portuguese nationality and a lower population density.

The implementation of commissioning processes was demonstrated to attenuate context-related discrepancies in 6.7% of the cases in cluster F1 and 3.3% in clusters F2 and F3. However, there is a widening of the discrepancies after commissioning in 3.3% of the cases in cluster F1, 16.7% in cluster F2, and 23.3% in cluster F3. These results show that the influence of the commissioning process on the results of the activity of PHC units could differ with different context characteristics [31].

Commissioning seems to be an important tool to attenuate context-related discrepancies in PHC performance results when these are related to “active population and employment” (F1 cluster) characteristics, but it is less effective when the discrepancies are related to nationality and population density. This suggests PHC teams can improve in adapting their care to patients of non-Portuguese nationality and work towards delivering culturally sensitive care [42]. However, it is also pertinent to note that there are fewer USFs in areas with a greater non-Portuguese population and in areas with a lower population density partly because, under the conditions at the time of the study, to implement an USF there was only required for a minimum number of users, making this more difficult to achieve in low-density population areas. This may also contribute to this result, as we know that commissioning is more effective in USF-type units (compared to the traditional USCP models). Nonetheless, primary healthcare services may be improved by, for example, embedding meaningful co-production approaches [43] into the commissioning process and involving patients of non-Portuguese nationality in the needs assessment and service planning and delivery stages of commissioning [44].

The evolution of the results of the nurse home visits indicator is statistically related to all the context factors analyzed. We emphasize the fact that it is an indicator in which health services meet people in a domestic setting. Therefore, this may reveal the way that they live, reducing social and cultural barriers and improving their health-related outcomes or care needs [45,46].

The indicators related to expenses of medicines and diagnostic tests, as well as the indicator of appropriate follow-up during the first year of life, did not show any increase in the differences in the results related to the context. This could mean that family doctors were not conditioned by commissioning in their prescriptions, regardless of the context in which they worked.

When analyzing cluster F3, which is related to nationality and population density, we see that there was an increase in discrepancies for the “proportion of patients with appropriate maternal health follow-up” in USF-A units. This may be due to the fact that many pregnant women of non-Portuguese nationality live abroad and only move to Portugal at a late stage of pregnancy [47]. In contrast, “appropriate follow-up in the first year of life”, which refers to child surveillance, shows no context-related discrepancies, which might be because there is already an established contact between the mother and the health unit, and teams are able to proactively invite newborn children to the Child Health Program consults.

## 5. Limitations and Suggestions for Future Research and Actions

This study has a few limitations that should be highlighted for the benefit of future research. The same target (control or number of consultations) was used for all patients, when health needs may often be different. It would also be relevant to analyze satisfaction and impact indicators.

The results of the KMO test show that it may be useful to deepen the investigation by including more variables.

The analysis is based on the results of indicators and deserves to be re-evaluated, taking into account its limitations [48].

For future studies and actions, we recommend exploring data per user regarding their use of services and health outcomes.

## 6. Conclusions

The trends observed in the results of the commissioning indicators are statistically related to the studied context factors, and the PHC unit organizational model is the most significant context factor in this regard. Therefore, it is important to encourage the implementation of a greater number of USFs, especially in low-density areas. Furthermore, regardless of the context, USFs have better results with lower costs for medicines and diagnostic tests.

The results for PHC indicators used in the commissioning processes were improved [17], and there was no evidence that the better results exhibited were achieved at the expense of a detrimental healthcare offered to vulnerable populations, but this was not associated with a reduction in healthcare inequalities. The differences in performance among PHC units in different contexts remained the same after the implementation of the commissioning process in 80% of the cases.

Commissioning processes should be adapted to better satisfy the needs of patients without Portuguese nationality and those living in low-density population areas. Further research could focus on understanding the challenges of delivering care among these populations.

## Figures and Tables

**Figure 1 ijerph-19-14819-f001:**
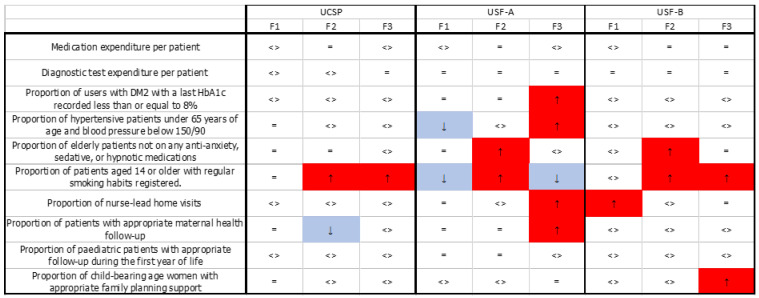
Summary of differences in results in distinct socio-economic contexts before and after commissioning by factors and indicators. Legend: <> differences before commissioning remain equal; = the absence of differences before commissioning remains; ↓ differences before commissioning decrease; ↑ differences before commissioning increase.

**Table 1 ijerph-19-14819-t001:** Description of commissioning indicators for Portuguese PHC units (2014–2016), socio-economic determinants analyzed and Portuguese PHC type of units.

Commissioning Indicators for Portuguese PHC Units (2014–2016)	Description
Medication expenditure per patient	Average total cost of subsidized medicines prescribed by PHC doctors per patient registered at the PHC unit.
Diagnostic test expenditure per patient	Average total cost of diagnostic test ordered by PHC doctors per patient registered at the PHC unit.
Proportion of users with DM2 with a last HbA1c recorded less than or equal to 8%	Proportion of DM patients whose last Hb1Ac measure was less than 8% (64 mmol/mol).
Proportion of hypertensive patients under 65 years of age with blood pressure below 150/90 mmHg	Proportion of hypertensive patients aged 65 years or above whose last blood pressure measurement was less than 150/90 mmHg.
Proportion of elderly patients not on any antianxiety, sedative or hypnotic medications	Proportion of patients aged 65 or older who were not prescribed antianxiety, sedative or hypnotic medications during the period of analysis in the past year.
Proportion of patients aged 14 or older with regular registered smoking habits	Proportion of patients aged 14 or older with regular smoking habits registered over the past 36 months.
Proportion of nurse-led home visits	Number of home visits carried out by PHC nurses per 1000 patients registered at the practice.
Proportion of patients with appropriate maternal health follow-up	Index accounting for the number of medical and nurse follow-up appointments, mandatory screening tests and diagnostic tests.
Proportion of pediatric patients with appropriate follow-up during the first year of life	Index accounting for the number of medical and nurse follow-up appointments, mandatory screening tests and completion of the national vaccination schedule in the first year of life.
Proportion of child-bearing-age women with appropriate family planning support	Index accounting for medical and nurse family planning appointments, as well as appropriate cervical cancer screening.
**Socio-economic Determinants**	**Description**
Percentage of elderly patients	Proportion of patients aged 65 years and older.
Percentage of users holding Portuguese nationality	Percentage of users with Portuguese nationality.
Percentage of users in economic deprivation	Households in which average monthly income, divided by the number of people responsible for the household, does not exceed 1.5 times the value of the indexing of social support (which, in 2019, was EUR 653.64).
Population density	The intensity of settlement expressed as the ratio between (total) population and surface (land) area (usually expressed as the number of inhabitants per square kilometer).
Unemployment rate	Rate that defines the relationship between the unemployed population and the labor force.
School dropout rate before mandatory education completed	Population aged between 10 and 15 years who dropped out of school without completing 9th grade—compulsory schooling.
**Organizational Models/Type of Unit**	**Description**
USF Model B	Self-organized group of professionals, with a practice’s financial incentives schemes linked to overall team achievement plus pay-for-group performance for each doctor, nurse and administrative staff group.
USF Model A	Self-organized group of professionals, with a practice’s financial incentive schemes linked to overall team achievement.
USCP	Traditional model without financial incentive schemes or pay-for-performance scheme.
USF Model C	Experimental model regulated by a special law that is not yet implemented, but is meant to complement eventual shortcomings in the National Health Service. Model C comprises USFs from the social, cooperative and private sectors in conjunction with the health center, but with no hierarchic dependency. Their activity is based on a contract signed with the regional health administration.

**Table 2 ijerph-19-14819-t002:** Relationship between context factors and evolution of the PHC indicator results (*p*-value).

	Type of Unit	Population Density (Average)	School Dropout Rate before Mandatory Education Completed (Average)	Unemployment Rate (Average)	Percentage of Elderly Patients (Average)	Percentage of Users in Economic Deprivation (Average)	Percentage of Users Holding Portuguese Nationality (Average)
Medication expenditure per patient	<0.01	0.52	0.78	0.01	<0.01	0.55	0.01
Diagnostic test expenditure per patient	<0.01	0.02	0.01	0.20	<0.01	0.88	0.20
Proportion of users with DM2 with a last HbA1c recorded less than or equal to 8%	<0.01	<0.01	0.27	0.29	0.68	0.85	<0.01
Proportion of hypertensive patients under 65 years of age with blood pressure below 150/90	<0.01	0.03	0.37	0.21	0.06	0.00	<0.01
Proportion of elderly patients not on any antianxiety, sedative or hypnotic medications	<0.01	0.24	0.96	0.00	0.33	0.93	<0.01
Proportion of patients aged 14 or older with regular registered smoking habits	<0.01	<0.01	0.09	0.01	0.96	0.00	<0.01
Proportion of nurse-led home visits	<0.01	<0.01	0.05	0.04	<0.01	0.01	<0.01
Proportion of patients with appropriate maternal health follow-up	<0.01	0.09	<0.01	0.02	<0.01	0.06	0.08
Proportion of pediatric patients with appropriate follow-up during the first year of life	<0.01	0.06	<0.01	<0.01	0.08	0.06	<0.01
Proportion of child-bearing-age women with appropriate family planning support	<0.01	0.02	0.67	0.92	0.01	0.01	<0.01

**Table 3 ijerph-19-14819-t003:** Results of context factors by PHC unit type (averages).

Type of Unit	Population Density	School Dropout Rate before Mandatory Education Completed	Unemployment Rate	Percentage of Elderly Patients	Percentage of Users in Economic Deprivation	Percentage of Users Holding Portuguese Nationality
UCSP	864.78	57.96	6.71	25.26	52.48	96.74
USF-A	1621.71	51.15	7.54	21.49	51.39	97.76
USF-B	1559.54	49.72	8.04	19.45	50.83	98.59
Average	1250.63	53.80	7.33	22.52	51.70	97.57

**Table 4 ijerph-19-14819-t004:** Indicator results (2016) by PHC unit type.

2016 Results	UCSP	USF-A	USF-B	Average
Medication expenditure per patient	195.07	162.72	137.54	168.94
Diagnostic test expenditure per patient	64.77	55.27	48.69	57.38
Proportion of users with DM2 with a last HbA1c recorded less than or equal to 8%	48.19	67.51	76.88	61.83
Proportion of hypertensive patients under 65 years of age with blood pressure below 150/90	37.55	58.43	72.51	53.66
Proportion of elderly patients not on any antianxiety, sedative or hypnotic medications	65.04	64.26	63.55	64.37
Proportion of patients aged 14 or older with regular registered smoking habits	47.18	69.75	77.61	62.05
Proportion of nurse-led home visits	155.7	134.17	146.82	148.49
Proportion of patients with appropriate maternal health follow-up	0.56	0.75	0.86	0.7
Proportion of pediatric patients with appropriate follow-up during the first year of life	0.73	0.86	0.93	0.83
Proportion of child-bearing-age women with appropriate family planning support	0.51	0.69	0.79	0.64

## Data Availability

The data presented in this study are available on request from the corresponding author.

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
