# Peer review of "Healthcare Equity and Commissioning: A Four-Year National Analysis of Portuguese Primary Healthcare Units"

_ijerph, 2022, doi:10.3390/ijerph192214819_

Round 1
Reviewer 1 Report
The paper deals with an extremely important theme for health policies and for the European and Portuguese health systems, on healthcare equity and commissioning, of Portuguese Primary Healthcare units.
Very well written paper. The title reflects the whole study and is very well elaborated to arouse more interest to the reader.
The introduction fits very well the problematic and with a connection to the objectives of the study. They approach an extremely current and pertinent problem in the current, difficult phase of the Portuguese National Health Service integrated in the European space, where it has an example to demonstrate. The authors present the organizational model of the USFs in the Portuguese health system, indicating the USFs model A and model B. These are the ones that are in operation. However, in my humble opinion, they could have referred to the overall organizational model of USFs and to the possibility of model C. These USFs of model C, which are very interesting and of great reformist value in the Portuguese health system and a European reference, could already be in force. The non-existence of model C USFs is only due to a lack of decision by health policies, which could have a great impact on the health outcomes of the Portuguese population, to improve the accessibility of citizens and reduce health inequalities. If there was a reformist spirit in health policies in Portugal, model C USFs would already be implemented. The non-implementation of the financial autonomy of the ACES, the maintenance of USF model B quotas and the absence of USF model C are a worrying limitation of the capacity for health reform in Portugal.
Chapter 2. Materials and Methods, summarized but clearly explicit of the methodology correctly used.
In table 1, the row "Percentage of users in economic deprivation" has a different font than the rest.
Chapters 3. Results and 4. Discussion very objective and well developed.
In line 191, they present a KMO=0.6, and could have made some reference to the impact of this value, in the conclusions of the study.
Chapter 5. Limitations and Suggestions for Future Research and Actions with a correct approach and pertinent suggestions for future work.
Set of bibliographical sources adequate to the type of study and with scientific actuality.
I offer to the authors my effusive congratulations for the paper.
Author Response
Dear Prof. Dr. Paul B. Tchounwou,
Thank you very much for the opportunity to revise our manuscript and for your constructive comments.
Please find below a point-by-point response to all comments.
Yours sincerely,
on behalf of all co-authors.
Point 1: The paper deals with an extremely important theme for health policies and for the European and Portuguese health systems, on healthcare equity and commissioning, of Portuguese Primary Healthcare units.
Very well written paper. The title reflects the whole study and is very well elaborated to arouse more interest to the reader.
The introduction fits very well the problematic and with a connection to the objectives of the study. They approach an extremely current and pertinent problem in the current, difficult phase of the Portuguese National Health Service integrated in the European space, where it has an example to demonstrate. The authors present the organizational model of the USFs in the Portuguese health system, indicating the USFs model A and model B. These are the ones that are in operation. However, in my humble opinion, they could have referred to the overall organizational model of USFs and to the possibility of model C. These USFs of model C, which are very interesting and of great reformist value in the Portuguese health system and a European reference, could already be in force. The non-existence of model C USFs is only due to a lack of decision by health policies, which could have a great impact on the health outcomes of the Portuguese population, to improve the accessibility of citizens and reduce health inequalities. If there was a reformist spirit in health policies in Portugal, model C USFs would already be implemented. The non-implementation of the financial autonomy of the ACES, the maintenance of USF model B quotas and the absence of USF model C are a worrying limitation of the capacity for health reform in Portugal.
Response 1:
We agree with this observation and have included additional information about model C USF in Table 1. However, a discussion about model C and the financial autonomy of the ACES would require a more extensive explanation of how these word and we considered that may be outside the scope of the article.
Point 2: Chapter 2. Materials and Methods, summarized but clearly explicit of the methodology correctly used.
Response 2:Thank you
Point 3: In table 1, the row "Percentage of users in economic deprivation" has a different font than the rest.
Response 3: Thank you for pointing this out.
Point 4: Chapters 3. Results and 4. Discussion very objective and well developed.
Response 4:Thank you
Point 5: In line 191, they present a KMO=0.6, and could have made some reference to the impact of this value, in the conclusions of the study.
Response 5: We agree with the reviewer and appreciate the suggestion. We included additional information about this limitation, please see section Limitations and Suggestions for Future Research and Actions.
Point 6: Chapter 5. Limitations and Suggestions for Future Research and Actions with a correct approach and pertinent suggestions for future work.
Response 6:Thank you
Point 7: Set of bibliographical sources adequate to the type of study and with scientific actuality.
Response 7:Thank you
Point 8: I offer to the authors my effusive congratulations for the paper.
Response 8: Thank you so much
Reviewer 2 Report
To whom it concerns,
Dear Madam/Sirs,
please find the comments in the attached file.

Author Response
Dear Prof. Dr. Paul B. Tchounwou,
Thank you very much for the opportunity to revise our manuscript and for yourconstructive comments.
Please find below a point-by-point response to all comments.
Yours sincerely,
on behalf of all co-authors
Point 1:
Article: highlighting areas of weakness, the testability of the hypothesis, methodological inaccuracies, missing controls, etc. As the author highlights the weaknesses in the Limitations section.
Please use the MeSH words in the keywords.
Response 1:. We agree with this observation and was included additional MeSH word in keywords
Point 2:
In the section 3.2. Differences in results in distinct socio-economic contexts before and after commissioning author is referring to the figure 3, but it does not exist
Response 2: Thank you for pointing this out.
Point 3:
Please refere to more recent publications.
Response 3: We agree with the reviewer and appreciate the suggestion. We included additional references:
- Watkins MW. Exploratory factor analysis: A guide to best practice. Journal of Black Psychology. 2018;44(3):219–46. Available online: https://doi.org/10.1177/0095798418771807 (accessed on 30/10/2022).
- Hart F. Is commissioning the enemy of co-production? Perspectives in Public Health. 2022;142(4):191-192. Available online: doi:10.1177/17579139221103189 (accessed on 30/10/2022).
- Elke Loeffler & Tony Bovaird (2019) Co-commissioning of public services and outcomes in the UK: Bringing co-production into the strategic commissioning cycle, Public Money & Management, 39:4, 241-252, Available online: DOI: 10.1080/09540962.2019.1592905 (accessed on 30/10/2022).
- Kruse FM, Ligtenberg WMR, Oerlemans AJM, Groenewoud S, Jeurissen PPT. How the logics of the market, bureaucracy, professionalism and care are reconciled in practice: An empirical ethics approach - BMC health services research [Internet]. BioMed Central. BioMed Central; 2020. Available online: https://bmchealthservres.biomedcentral.com/articles/10.1186/s12913-020-05870-7 (accessed on 30/10/2022).
- Santana P, Almendra R. The health of the Portuguese over the last four decades. Méditerranée. 2018;(130). Available online: https://doi.org/10.4000/mediterranee.10348 (accessed on 30/10/2022).
- Macfarlane AJR. What is clinical governance? BJA Education. 2019;19(6):174–5. Available online: https://doi.org/10.1016/j.bjae.2019.02.003 (accessed on 30/10/2022).
Point 4:
In the section 3.2. Differences in results in distinct socio-economic contexts before and after commissioning author is referring to the figure 3, but it does not exist
Response 4: Thank you for pointing this out. We apologize for the mistake.
Reviewer 3 Report
The topic of this paper is highly relevant. We know very little about the impact of different organisational set-ups on equity and other parameters. However, the paper requires some more effort. I assume that the authors have all materials, but they should also present it.
Abstract: I had strong problems to understand the abstract. Without knowing the content of the paper, it is very difficult to read and understand. I assume the abstract must be rewritten completely
PHC: Throughout the paper, you use the words "primary care" and "primary healthcare" as synonyms. From my understanding, "primary care" is a level of the health care system while "primary healthcare" acc. to WHO (e.g. Alma Ata declaration) is a concept or philosophy of healthcare provision. Safeguard that the right term is used. And if the Portuguese Government mixes up terms, reflect on it, at least in the discussion section.
It might be helpful to present a table with the differences between the respective models.
It might be helpful to base your discussion on organisational theories. Would be interesting to see which one applies best.
Table 1 is very big and holds a lot of information. Most of it is not sufficiently described in the text.
You talk about “descriptive statistics” but present p-values?
The method “factorial analysis” can mean many things. You should describe mathematically what you did.
Table 2 does not really help. The normal presentation would be to show the real value, not only the significance
The discussion is disappointing. I think there is a wealth of literature on inequalities to which you could refer.
Line 55: “poor mental health” or “mental health problems”.
Line 77: “[78]” instead of “(18]”
English must be improved.
Author Response
Dear Prof. Dr. Paul B. Tchounwou,
Thank you very much for the opportunity to revise our manuscriptfor your constructive comments.
Please find below a point-by-point response to all comments.
Yours sincerely,
on behalf of all co-authors.
Point 1: The topic of this paper is highly relevant. We know very little about the impact of different organisational set-ups on equity and other parameters. However, the paper requires some more effort. I assume that the authors have all materials, but they should also present it.
Response 1: We have introduced further explanations and references on this topic. We hope that it has become clearer and more complete.
Point 2: Abstract: I had strong problems to understand the abstract. Without knowing the content of the paper, it is very difficult to read and understand. I assume the abstract must be rewritten completely
Response 2:
The abstract was structured and rewritten in order to make it easier to understand the paper.
Point 3: PHC: Throughout the paper, you use the words "primary care" and "primary healthcare" as synonyms. From my understanding, "primary care" is a level of the health care system while "primary healthcare" acc. to WHO (e.g. Alma Ata declaration) is a concept or philosophy of healthcare provision. Safeguard that the right term is used. And if the Portuguese Government mixes up terms, reflect on it, at least in the discussion section.
Response 3:
Thank you for pointing this out. We reviewed the terms, considering the scope of care and the concept used by the WHO and also this bibliography
https://pubmed.ncbi.nlm.nih.gov/17120883/
https://primarycare.imedpub.com/what-is-the-difference-between-primary-care-and-primary-healthcare.pdf
Point 4: It might be helpful to present a table with the differences between the respective models.
Response 4: Table 1 was duly updated.
Point 5: It might be helpful to base your discussion on organisational theories. Would be interesting to see which one applies best.
Response 5: We agree with the reviewer and appreciate the suggestion. We included additional information about organizational theories within the distinct existing organizational models.
Point 6: Table 1 is very big and holds a lot of information. Most of it is not sufficiently described in the text.
Response 6: We agree with the reviewer, we tried improve the description.
Point 7: You talk about “descriptive statistics” but present p-values?
Response 7: Thank you for pointing this out. Generalized estimating equation model was used to analyze whether the evolution of commissioning indicators is related to the PHC units’ socio-economic context variables. Additional information was included in section Materials and Methods.
Point 8: The method “factorial analysis” can mean many things. You should describe mathematically what you did.
Response 8: Additional information was included in section Materials and Methods.
Point 9: Table 2 does not really help. The normal presentation would be to show the real value, not only the significance
Response 9: Thank you for pointing this out. Table 2 presents the relation between context factors and evolution of the PHC indicator results. P-values were calculated considereing the evolution of the commissioning indicator result (a two-point analysis - 2013 and 2016) and it relation with each of six socio-economic determinants or the type of unit using the generalized estimating equation model. Our option for using the GEE was due to the robustness of the model and the ability it has to use the information that was collected (result of the indicators, of each unit, for each of the year under analysis).
Point 10: The discussion is disappointing. I think there is a wealth of literature on inequalities to which you could refer.
Response 10: We thank you for the feedback. We included an additional discussion points on ways the services can improve, for example, through co-production and participate service design approaches. We hope this enriches the discussion.
Point 11: Line 55: “poor mental health” or “mental health problems”.
Response 11 Thank you for pointing this out.
Point 12: Line 77: “[78]” instead of “(18]”
Response 12 Thank you for pointing this out.
Point 13: English must be improved.
Response 13: A detailed language check (spelling, grammar, sentence structures and terminology) was made by MDPI Language Editing Services.
Round 2
Reviewer 3 Report
The authors have addressed the issues raised. I merely recommend changing the sentence "The PHC reforms in Portugal can be also analyzed by organizational theories". Theories do not analyse. But you could write: "The PHC reforms in Portugal can be also analyzed within their organizational framework".
Author Response
Dear Prof. Dr. Paul B. Tchounwou,
Thank you very much for the opportunity to revise our manuscript and for yourconstructive comments.
Please find below a point-by-point response to all comments.
Yours sincerely,
on behalf of all co-authors
Point 1: The authors have addressed the issues raised. I merely recommend changing the sentence "The PHC reforms in Portugal can be also analyzed by organizational theories". Theories do not analyse. But you could write: "The PHC reforms in Portugal can be also analyzed within their organizational framework".
Response 1: We agree with the reviewer and included the suggestion.
